# Vaginal Dysbiosis in Infertility: A Comparative Analysis Between Women with Primary and Secondary Infertility

**DOI:** 10.3390/microorganisms13010188

**Published:** 2025-01-17

**Authors:** Iliana Alejandra Cortés-Ortíz, Gustavo Acosta-Altamirano, Rafael Nambo-Venegas, Jesús Alejandro Pineda-Migranas, Oscar Giovanni Ríos-Hernández, Eduardo García-Moncada, Alejandra Yareth Bonilla-Cortés, Mónica Sierra-Martínez, Juan Carlos Bravata-Alcántara

**Affiliations:** 1Laboratorio de Genética y Diagnóstico Molecular, Hospital Juárez de México, Instituto Politécnico Nacional, 5160, Col. Magdalena de las Salinas, Ciudad de México 07760, Mexico; iliancortes@yahoo.com.mx (I.A.C.-O.); jesuspm23@yahoo.com.mx (J.A.P.-M.); ogiovannirh311001@gmail.com (O.G.R.-H.); eduardo.garcia.moncada@gmail.com (E.G.-M.); 2Hospital General de México, Eje 2A Sur (Dr. Balmis) No.148, Cuauhtémoc, Doctores, Ciudad de México 06726, Mexico; mq9903@live.com.mx; 3Laboratorio de Genómica del Envejecimiento, Centro de Investigación sobre Envejecimiento (CIE-CINVESTAV Sede Sur), Instituto Nacional de Medicina Genómica (INMEGEN), Ciudad de México 14330, Mexico; rafaelnambo@yahoo.com.mx; 4Facultad de Estudios Superiores Cuautitlán, Universidad Nacional Autónoma de México (UNAM), Av. 1º de Mayo S/N, Santa María las Torres, Campo Uno, Cuautitlán Izcalli 54740, Mexico; 5Escuela Nacional de Medicina y Homeopatía, Instituto Politécnico Nacional, Av. Guillermo Massieu Helguera 239, La Purísima Ticomán, Gustavo A. Madero, Ciudad de México 07320, Mexico; yarethcortes1998@gmail.com; 6Unidad de Investigación en Salud, Hospital de Alta Especialidad Ixtapaluca, IMSS-Bienestar, Carr Federal México-Puebla Km 34.5, Ixtapaluca 56530, Mexico

**Keywords:** infertility, vaginal microbiota, HPV

## Abstract

Infertility, both primary and secondary, is strongly influenced by microbiological factors, with the vaginal microbiota playing a key role in reproductive health. Objective: The aim of this study was to characterize the vaginal microbiota of 136 Mexican women diagnosed with infertility—primary (n = 58) and secondary (n = 78)—by evaluating the presence of pathogenic bacterial species and their associations with infertility conditions. Methods: Samples were obtained through cervical swabs, and microorganism identification was performed using qPCR techniques. Results: Analysis revealed a positive correlation between increased age and the likelihood of primary infertility, as well as a negative correlation with secondary infertility. Significant differences in microbial composition were also observed between the two infertility groups. *Lactobacillus crispatus* and *Lactobacillus gasseri* were dominant in women with primary infertility, in addition to a high prevalence of *Gardnerella vaginalis* and *Fannyhessea vaginae*. Additionally, correlations were found between the presence of human papillomavirus (HPV) and sexually transmitted bacteria, as well as *Gardnerella vaginalis*. Conclusion: Our findings suggest that the composition of the vaginal microbiota may play a decisive role in infertility, highlighting the need for personalized therapeutic strategies based on microbial profiles.

## 1. Introduction

Infertility is a global issue affecting approximately 15% of couples of reproductive age, and it significantly impacts the physical, mental, and social well-being of those affected [1]. Recent research has shown that socioeconomic factors, such as limited access to healthcare, can exacerbate the prevalence of primary infertility in certain populations, especially in those with limited resources [2].

Among the causes of female infertility, primary infertility (the inability to conceive after 12 months of trying) and secondary infertility (difficulty achieving a new pregnancy after a previous one) represent two distinct clinical categories with different etiological factors; primary infertility is often linked to anatomical or genetic abnormalities, such as blocked fallopian tubes or chromosomal issues, while secondary infertility is more frequently associated with factors like infections or charges in the vaginal microbiota, which can lead to conditions such as bacterial vaginosis or pelvic inflammatory disease [3]. While hormonal, anatomical, and genetic factors have been extensively studied as primary causes of infertility, there has been growing interest in the role of the vaginal microbiota in reproductive health in recent years [4].

The vaginal microbiota, predominantly composed of *Lactobacillus* species in healthy women, is essential for maintaining an acidic environment in the reproductive tract, which protects against the colonization of opportunistic pathogens [5]. The microbiota can be classified into five community state types (CST) based on the dominance of specific *Lactobacillus* species: CST 1 is dominated by *Lactobacillus crispatus*, CST II by *Lactobacillus gasseri*, CST III by *Lactobacillus iners*, CST IV by anaerobic bacteria, and CST V by *Lactobacillus jensenii*. A balanced vaginal microbiota helps to prevent infections such as bacterial vaginosis, which can disrupt the vaginal microenvironment and negatively affect fertility [6]. Female infertility can result from a variety of causes, including hormonal imbalances, anatomical abnormalities, and conditions such as polycystic ovary syndrome (PCOS) and endometriosis [7]. These factors not only directly impact reproductive health, but also influence the vaginal microbiota. For instance, hormonal fluctuations can alter the composition of the vaginal microbiota, promoting the overgrowth of anaerobic bacteria and reducing the prevalence of beneficial *Lactobacillus* species.

Studies have shown that different causes of infertility are associated with distinct CSTs. CST IV, characterized by a low abundance of *Lactobacillus* and high microbial diversity, has been linked to poorer reproductive outcomes [7]. Recent studies have emphasized that vaginal dysbiosis (i.e., an imbalance in dominant microbial communities) is associated with a higher risk of both primary and secondary infertility [8]. Furthermore, alterations in the vaginal microbiota have been correlated with cervical cytological changes, including the presence of human papillomavirus (HPV) infections, which are highly prevalent in certain populations and can influence reproductive health outcomes [9].

Various studies have explored the relationships between the composition of the vaginal microbiota and infertility, revealing that the loss of *Lactobacillus* species, especially *Lactobacillus crispatus*, is associated with an increased prevalence of infections caused by *Gardnerella vaginalis* and *Fannyhessea vaginae*—bacteria linked to bacterial vaginosis [10,11]. These microbial alterations are also associated with an increased susceptibility to sexually transmitted infections (STIs), such as HPV, which can further impair reproductive capacity [12]. Additionally, studies indicate that in populations such as Hispanic women, the diversity of the vaginal microbiota, with a predominance of *Lactobacillus iners* and anaerobic bacteria, is associated with an increased risk of dysbiosis, which negatively impacts reproductive outcomes [9]. However, most studies have focused on populations from developed countries, leaving a significant gap in understanding how these microbial dynamics impact women in different sociocultural contexts, such as those in Mexico [13].

Among Mexican women, primary infertility is more common than secondary infertility, contrasting with global reports where secondary infertility tends to be more prevalent [14]. This difference may be related to the high prevalence of inadequately treated infections and other socioeconomic factors that impact reproductive health [15]. Given that the vaginal microbiota can vary significantly depending on geographic, ethnic, and environmental factors, conducting specific studies in Mexican populations is essential to understand the unique characteristics of their microbiota and how these may influence fertility [16].

While vaginal dysbiosis has been linked to infertility, there is little information on the differences in the vaginal microbiota between women with primary and secondary infertility [17]. Understanding these differences is crucial for developing personalized clinical interventions that could improve conception rates in these patients [18]. This study aimed to characterize the dysbiotic vaginal microbiota and its relationship with infertility conditions in women diagnosed with primary and secondary infertility.

## 2. Materials and Methods

### 2.1. Population

This study included Mexican women of reproductive age who visited the Gynecology Department at Hospital Regional de Alta Especialidad de Ixtapaluca, IMSS-Bienestar, between January 2023 and May 2024. A total of 136 women diagnosed with infertility participated in the study, of whom 58 presented with primary infertility and 78 with secondary infertility. Clinical information for each participant was obtained following the hospital’s standard procedures. The study was approved by the Ethics Committee of Hospital Regional de Alta Especialidad de Ixtapaluca, IMSS-Bienestar (NR-105-2023).

Samples were collected before the participants’ menstrual period; participants reported abstaining from sexual intercourse for five days prior and had not received antibiotic treatment in the past month. Sample collection was performed by gynecologists during clinical consultations, obtaining vaginal secretions from the lower third of the vagina (CL), the posterior fornix (CU), and cervical mucus from the cervical canal (CV) to assess the microbiota in different anatomical regions. Samples were collected using Classic Swabs (Copan), with 3 to 4 swabs in the epithelium of each area under direct visualization. The swabs from all regions were placed in one BD Universal Viral Transport system (Becton, Dickinson and Company, Sparks, MD, USA) to detect viruses, *Chlamydia*, *Mycoplasma*, and *Ureaplasma*. Subsequently, the swabs were then placed in sterile 1.5 mL tubes (Eppendorf, Hamburg, Germany) and immediately frozen at −80 °C. Total DNA was obtained from 400 µL of each patient’s cervical swab. Extraction and purification were performed using a commercial kit (Invitek Molecular GmbH, Berlin, Germany), following the manufacturer’s instructions. The extracted DNA was stored at −20 °C until analysis by qPCR.

### 2.2. Molecular Detection of Lactobacillus, Sexually Transmitted Bacterial, and Bacteria Associated with Bacterial Vaginosis

To detect clinically relevant bacteria, specific primers and probes for qPCR were designed using the Primer3plus software version: 3.3.0. The analyzed species included *Lactobacillus crispatus*, *Lactobacillus jensenii*, *Lactobacillus gasseri*, *Lactobacillus iners*, *Gardnerella vaginalis*, *Fannyhessea vaginae*, and *Mobiluncus mulieris*, as detailed in Appendix A. The specificity of the molecular markers was evaluated through in silico analysis. Amplification was performed using a PCR Master Mix kit (Luna Universal Probe qPCR Master Mix, M3004L, New England BioLabs Inc., Ipswich, MA, USA), with a concentration of 10 pM for each primer (forward and reverse), 1 pM for the probe, and 1 µg of DNA as a template. PCR was conducted on a CFX96 Real-Time system (Bio-Rad, Hercules, CA, USA) under the following conditions: initial denaturation at 95 °C for 3 min; denaturation at 95 °C for 10 s; and annealing and extension at 58 °C for 30 s, for a total of 39 cycles.

For sexually transmitted bacteria, the STI-7 Detection kit (v1.1) (Anyplex™ II, SD7700Y, MT Promedt Consulting GmbH, St. Ingbert, Germany) was used, which enables the identification of *Neisseria gonorrhoeae* (NG), *Chlamydia trachomatis* (CT), *Mycoplasma genitalium* (MG), *Mycoplasma hominis* (MH), *Ureaplasma urealyticum* (UU), *Ureaplasma parvum* (UP), and *Trichomonas vaginalis* (TV). All procedures were performed according to the manufacturer’s specifications.

### 2.3. Viral Detection and Other Micro-Organisms

Considering that the microbiota includes not only bacteria but also viruses, fungi, and other micro-organisms, we analyzed viruses relevant to infertility and reproductive health. HPV detection was performed using the HPV28 Detection kit (Anyplex™ II, HP7S00X, MT Promedt Consulting GmbH, St. Ingbert, Germany), while herpes virus and bacteria detection was carried out with the MeningoFinder 2Smart kit (PathoFinder, Maastricht, The Netherlands). The HPV28 kit allowed for the detection of 19 high-risk HPV genotypes (types 16, 18, 26, 31, 33, 35, 39, 45, 51, 52, 53, 56, 58, 59, 66, 68, 69, 73, and 82) and 9 low-risk genotypes (types 6, 11, 40, 42, 43, 44, 54, 61, and 70). The MeningoFinder 2Smart kit identified a range of viral and bacterial agents, including Herpes simplex virus type 1; Varicella-zoster virus, Epstein–Barr virus, Cytomegalovirus, Human Herpesvirus types 6, 7, and 8, *Staphylococcus aureus*, *Haemophilus influenzae*, *Streptococcus agalactiae*, *Escherichia coli* K1, *Cryptococcus gattii*, and *Cryptococcus neoformans*. All procedures were performed according to the manufacturer’s specifications.

### 2.4. Statistical Analysis

Logistic regression analysis was performed to determine the associations of the microbiota with each infertility group. Significance was determined for *p*-values less than 0.05. Training and test groups were created through random sampling, and the ratio between them was adjusted to optimize the model’s predictive capacity.

To compare microbial communities by community state type, we carried out the PERMANOVA test (permutational multivariate analysis of variance), implemented with the Vegan 2.6-4 package in R. The obtained F value was compared with values from 999 random permutations across groups, with statistical significance considered for *p*-values < 0.05. Both weighted and unweighted unifrac distances were used to evaluate the differences between communities.

Additionally, we performed principal component analysis (PCA), clustering using the K-Means method, and Random Forest analysis to identify the variables with the highest impacts on infertility groups. All statistical and data visualization analyses were conducted in RStudio 2024, using R version 4.4.0 and the Tidyverse package version 2.0.0. (Posit, Boston, MA, USA).

## 3. Results

### 3.1. Population

A total of 136 Mexican patients diagnosed with infertility were included, of whom 58 (43%) presented with primary infertility and 78 (58%) with secondary infertility. Patients with primary infertility represented a smaller proportion of the cohort compared to those with secondary infertility. These groups differed not only in prevalence but also in the distribution of their clinical presentations. All patients were referred from primary healthcare centers with a diagnosis of infertility, and only those with cervical and tubal factors (obstruction of the fallopian tubes) were included. Patients with endocrine or chromosomal factors, as well as cases where male factor infertility was identified, were excluded from the study.

The age distribution of patients with primary and secondary infertility ranges from 18 to 47 years (Figure 1). For primary infertility, there is a higher concentration of patients at 37 years, with a slight bias toward older ages (between 25 and 40 years). In contrast, patients with secondary infertility exhibit a more uniform distribution, peaking at 29 years and showing greater representation in the 25–35 year range, extending up to 45 years. There is an overlap between both groups in the 25–40 year range, although patients with primary infertility tend to be older compared to those with secondary infertility. Overall, both types of infertility are underrepresented at the extremes of the age range.

### 3.2. Identification of Groups with Different Microbial Profiles

An exploratory cluster analysis using the K-Means method was conducted to identify groups with differentiated microbial profiles, utilizing all available variables. This analysis allowed for the identification of five well-defined clusters (Figure 2A). Based on this information, a Random Forest analysis was performed, which indicated that the variables with the greatest impact on group classification were age, community state type, presence of sexually transmitted bacteria, type of infertility, and presence of human papillomavirus (HPV) (Figure 2B).

### 3.3. Relationship Between Age and Type of Infertility

Age was identified as the most prominent variable in the Random Forest analysis. To evaluate its impact on the type of infertility, a logistic regression model was used with a training set with a ratio of 0.75 (i.e., 75% of the data) through random sampling. The logistic regression models revealed a significant relationship between age and primary infertility, showing a positive correlation (Figure 3); in contrast, secondary infertility presented a negative relationship with age. The results obtained are detailed in Appendix A.

### 3.4. Association Between Community State Type and Dominant Species

In both infertility types, communities dominated by more than one *Lactobacillus* species were found. The dominance of each CST was assigned according to the species detected in the highest concentration. To identify association patterns between different community state types (CSTs) and infertility, PERMANOVA models were conducted. Significant differences were found concerning CSTs (PERMANOVA (*p* = 0.001). The results obtained are detailed in Appendix A.

To determine the type of infertility associated with CSTs, a Multiple Logistic Regression analysis was performed. Only primary infertility significantly correlated with CSTs (*p* = 0.04748), predominantly CST I, CST II, and CST IV. In the secondary infertility group, *L. gasseri*, *L. crispatus*, and *L. iners* were observed; however, no statistically significant relationship was found between them.

### 3.5. Correlations Between Bacterial Species and Viruses in Different Community States

Principal component analysis (PCA) was used to evaluate the influences of various bacterial and viral species within each CST. In CST I, *Lactobacillus crispatus*, *Lactobacillus jensenii*, and *Lactobacillus gasseri* were the most influential species, showing positive correlations. In CST II, *Haemophilus influenzae* and *Streptococcus agalactiae* were the most influential bacteria, with no evidence of correlation between them. In CST III, sexually transmitted bacteria and herpes viruses (HHV7 and HHV6), as well as the Epstein–Barr virus, showed a positive correlation. In CST IV, positive correlations were observed between *Gardnerella vaginalis*, *Fannyhessea vaginae*, HPV, and *H. influenzae*, as indicated by the small angles between these variables (Figure 4).

HPV infections were present in 33% of cases of primary infertility and 26% of cases of secondary infertility, with a prevalence approaching statistical significance in women with primary infertility (*p* = 0.08651, Multiple Logistic Regression). The most frequent HR-HPV genotypes were 45 and 66 (21% in both cases), followed by 59 (10%), while the most common low-risk genotypes were 61 and 70 (10% in both cases). The results obtained are detailed in Appendix A.

Several clinically relevant correlation and coinfection analyses were performed. A linear regression correlation analysis between HPV and sexually transmitted bacteria showed a Pr(>|z|) value of 0.0383. When analyzing *Gardnerella vaginalis* relative to other bacteria, it was found to have a positive correlation with *Fannyhessea vaginae* (Pr(>|z|) = 0.00177), and it was also correlated with sexually transmitted bacteria (Pr(>|z|) = 0.02989). Additionally, Epstein–Barr virus (Pr(>|z|) = 0.05541) and *H. influenzae* (Pr(>|z|) = 0.06194) revealed values close to significance. The results obtained are detailed in Appendix A.

Using Multiple Logistic Regression, only *Gardnerella vaginalis* (Pr(>|z|) = 0.0248) showed a significant association with primary infertility, while HPV (Pr(>|z|) = 0.08651) and *Fannyhessea vaginae* (Pr(>|z|) = 0.07386) were close to significance, as shown in Appendix A.

In secondary infertility, *Gardnerella vaginalis* (Pr(>|z|) = 0.0123), sexually transmitted bacteria (Pr(>|z|) = 0.02989), exhibited a significant association, with Epstein–Barr virus (Pr(>|z|) = 0.05541) and *Haemophilus influenzae* (Pr(>|z|) = 0.00.06194) being close to statistical significance. The sexually transmitted bacteria identified were *Ureaplasma parvum* (84%), *Ureaplasma urealyticum* (25%), *Mycoplasma hominis* (11.3%), and *Chlamydia trachomatis* (2.2%), as shown in Appendix A.

## 4. Discussion

The microbiome is defined as the community of commensal, symbiotic, and pathogenic microorganisms inhabiting the body or a specific environment. It plays a critical role in both healthy development and the onset of various diseases [19]. In the female reproductive tract, the microbiome significantly influences conception, pregnancy outcomes, gynecological health, and fetal development [20]. In healthy women of reproductive age, *Lactobacillus* spp. predominate, with some fluctuations in abundance related to the phases of the menstrual cycle, including the follicular, ovulation, and luteal phases [21]. Determining the relationships between advanced age and the composition of the vaginal microbiota is crucial for understanding how female fertility is impacted over time and how this affects infertility treatments. With aging, changes in the vaginal microbiota—including a reduction in protective *Lactobacillus* prevalence and increased susceptibility to microbial imbalances or dysbiosis—may emerge [22]. These changes can lead to difficulties in natural conception and negatively impact assisted reproductive technology (ART) outcomes, which are generally less effective in older women [23,24]. Our findings align with this phenomenon, showing that older patients have a higher probability of primary infertility, while younger patients are more associated with secondary infertility. Hong et al., 2020 highlighted those various etiologies, including anatomical and chromosomal abnormalities, that are more prevalent in primary infertility. In contrast, secondary infertility often has a stronger microbiological component, involving imbalances in the vaginal microbiota [7].

Vargas-Robles et al. found that the prevalence of *Lactobacillus iners* and other anaerobic bacteria, which are less protective, was higher in Hispanic populations, suggesting that unique dysbiosis patterns might contribute to different infertility risks in these. Studies have suggested that modulating the microbiota and restoring a healthy microbial profile could improve ART outcomes in older women, emphasizing the need for personalized approaches to optimize reproductive environments according to age [25]. In this context, the effective management of the vaginal microbiota could become a key element in fertility protocols [26]. In our study, significant variability was observed in the composition of the vaginal microbiota among Mexican women with primary and secondary infertility, with distinct pathogen prevalences and a reduction in *Lactobacillus* spp. in primary infertility patients.

Compared to studies in European or Asian populations [27,28,29,30], our findings suggest that the dysbiosis characteristics in Mexican women may present unique features. Hernández-Rosas reported that CST IV, characterized by low *Lactobacillus* abundance and an increased diversity of anaerobic bacteria, is more likely to be associated with adverse reproductive outcomes. However, this observation does not align with our findings in the Mexican cohort [31]. The variability observed in bacterial profiles could be influenced by specific environmental, genetic, and social factors within the Mexican population, such as diet, hygiene practices, and access to healthcare services [32]. We identified specific patterns in women with infertility, suggesting that a detailed microbiological characterization could provide deeper insights into the mechanisms associated with infertility.

Previous studies have indicated that a decrease in beneficial species, such as *Lactobacillus crispatus*, and an increase in opportunistic pathogens can heighten susceptibility to infections and inflammation, potentially compromising endometrial receptivity and embryo implantation [33]. Our findings support this theory, showing that women with infertility have a more diverse vaginal microbiota that is less dominated by *Lactobacillus* spp. when compared to fertile women. In our study, the vaginal microbiota showed a notable influence on the type of microbial community present, which was distinctly associated with the type of infertility, namely primary or secondary. Additionally, the role of viral infections as cofactors in dysbiosis was analyzed in more detail. Coinfections involving HPV and herpes viruses may exacerbate microbial imbalances, negatively impacting fertility. This suggests that multidisciplinary studies integrating bacterial, viral, and fungal analysis could provide a more comprehensive understanding of the microbiome’s impact on female infertility [9]. In women with primary infertility, we identified significant associations between *Lactobacillus crispatus*, *Lactobacillus gasseri*, *Gardnerella vaginalis*, *Fannyhessea vaginae*, and HPV, which reflected the presence of CST I, CST II, and CST IV. Clinically, these findings suggest that the characterization of the vaginal microbiota could serve as a diagnostic tool to assess infertility risk in women of reproductive age. The interaction between the vaginal microbiota and the local immune system is crucial for endometrial receptivity. Alterations in the microbial composition can trigger immune responses that create an inflammatory environment, thereby negatively impacting embryo implantation and overall fertility [34,35,36]. Research on the role of the microbiota in ART outcomes has highlighted that a vaginal community dominated by *Lactobacillus* spp. (known as community state type or CST I) could enhance ART outcomes, due to their ability to maintain low pH levels and prevent the overgrowth of pathogens [37]. However, the prevalence of mixed communities with high microbial diversity—such as CST IV—has been associated with adverse outcomes, including implantation failure and spontaneous abortion, consistent with the microbial profile observed in our secondary infertility group. These findings reinforce the hypothesis that a balanced microbial environment is essential for endometrial receptivity and successful embryo implantation.

The relationships between age, microbiota changes, and fertility have also been widely discussed in the literature [38]. With age, the diversity and composition of the vaginal microbiome tend to vary, potentially destabilizing *Lactobacillus*-dominated CSTs and fostering dysbiotic conditions associated with infertility [39]. This phenomenon, observed in older women, could explain the higher prevalence of primary infertility within this age group in our study. Recent studies have also shown that supplementation with specific strains of probiotics, particularly *Lactobacillus crispatus*, may help restore the vaginal microbiota balance in women experiencing infertility, potentially improving ART success rates [40]. The literature also suggests that reduced *Lactobacillus* levels and increased pathogens, such as *Gardnerella vaginalis* and *Fannyhessea vaginae*, predispose individuals to conditions such as bacterial vaginosis, which may hinder conception in older women and affect the success of ART in this demographic [41].

The findings from our study align with previous reports indicating that pathogens such as *Ureaplasma parvum* and *Ureaplasma urealyticum* are frequently present in women with infertility [40], suggesting that these bacteria may play key roles in the pathophysiology of infertility within this population. *Mycoplasma hominis* was therefore not representative, similar to the findings reported by Bustos-López et al., who found no association between *Mycoplasma* spp. and any infertility or adverse pregnancy conditions in Mexican women [42]. Previous research has linked these pathogens to conditions disrupting endometrial function and embryo implantation receptivity [43]. The high prevalence of these pathogenic species in our infertility patients could indicate a direct link between dysbiosis and increased local inflammation.

Additionally, *G. vaginalis* and *F. vaginae* were identified as the bacteria most significantly associated with primary infertility. Ferris et al. reported that *G. vaginalis* alone may not be pathogenic; however, when coexisting with *F. vaginae*, bacterial vaginosis can occur. This occurs because *F. vaginae* initiates the colonization of the vaginal epithelium and acts as a scaffold for other species to adhere [44]. In our findings, *G. vaginalis* was also associated with HPV, which aligns with previous research [45,46,47] showing that *G. vaginalis* is strongly interconnected with HPV acquisition. Furthermore, greater diversity in vaginal microbiota significantly increases the risk of acquiring this virus [48]. Supporting these findings, Wee et al. identified a higher prevalence of *Gardnerella* in infertile women in conjunction with *Ureaplasma* [49].

In the same way, identifying viruses such as HPV in these patients raises questions about potential co-infection and its synergistic impact on the vaginal ecosystem [50]. In the case of HPV, the high-risk genotypes identified were consistent with those reported in two Mexican studies [32,51], suggesting a regional pattern of genotype prevalence that may be influenced by sociodemographic or behavioral factors. This concordance highlights the importance of targeted public health strategies, including vaccination and early screening programs, to mitigate the potential contribution of high-risk HPV genotypes to infertility.

The absence of *Chlamydia trachomatis* and the lack of dominance of *Lactobacillus iners* in the microbiota of the women evaluated in our study suggest significant differences in the microbiological and epidemiological characteristics of our population compared to those reported by Chen et al., who were Chinese women diagnosed with tubal infertility and with no history of common sexually transmitted infections. These findings underscore the need to consider population-specific factors, such as geographic environment, health behaviors, and analytical methods, when interpreting the dynamics of the vaginal microbiota and its association with gynecological conditions such as tubal infertility [52].

Our results suggest that bacterial and viral co-infections could exacerbate dysbiosis and contribute cumulatively to infertility. This hypothesis highlights the need for multidisciplinary studies that include bacterial, viral, and fungal evaluations to understand the microbiome’s role in female infertility fully.

Although this study provides valuable insights into the relationships between vaginal dysbiosis and infertility in a sample of Mexican women, some limitations should be addressed in future research. A key limitation of this study is its cross-sectional design, which prevents the establishment of causal relationships between dysbiosis and infertility. Additionally, the sample size may restrict the generalizability of the findings. Larger studies could provide a more comprehensive understanding of microbiological diversity across different infertility subgroups. Future research should consider longitudinal studies to clarify whether dysbiosis precedes infertility or arises as a consequence of other reproductive health disorders and to provide a more detailed characterization of microbiological diversity. Although qPCR and cluster analyses are robust tools, metagenomic sequencing techniques could offer a more complete view of the microbiota. In this sense, a multiomic analysis is recommended for the exhaustive characterization of the vaginal microbiome in patients with different infertility types. Future research should also explore the interactions between genetic factors, lifestyle, and microbiome alterations to understand the holistic influence on fertility, particularly in populations with unique environmental and genetic backgrounds [53].

Our findings underscore the clinical relevance of the vaginal microbiota in female infertility, indicating that dysbiosis could be a determinant of reproductive capacity. Identifying specific microbial patterns in women with primary and secondary infertility provides a solid basis to consider the vaginal microbiota as a therapeutic target in infertility treatments. In this context, implementing routine microbiological analyses in women struggling to conceive could help to identify those at risk of infertility due to dysbiosis, enabling early interventions such as probiotics or targeted antimicrobial therapies.

Furthermore, this study opens new research avenues to explore the complex interactions between bacteria, viruses, and the immune system within the reproductive microenvironment. The hypothesis of bacterial and viral co-infection in dysbiosis offers a novel perspective, suggesting that infertility may result from multiple microbiological factors, rather than being exclusively bacterial.

## 5. Conclusions

The results of this study underscore the fundamental role of the vaginal microbiota in female fertility, demonstrating that microbial imbalances can adversely impact the ability to conceive. Women’s infertility can stem from various causes, and understanding these causes is crucial for effective treatment. Our findings suggest that the characterization of specific microbial profiles based on the type of infertility may open up opportunities for personalized therapeutic interventions. In this regard, treatments involving probiotics or targeted antimicrobials could help to restore an optimal reproductive environment. Furthermore, the observed relationships between age and alterations in the microbiota highlight the importance of managing dysbiosis in older women, who face greater barriers to conception. Overall, these results indicate that the vaginal microbiota should be considered as a key factor in the evaluation and management of female infertility.

## Figures and Tables

**Figure 1 microorganisms-13-00188-f001:**
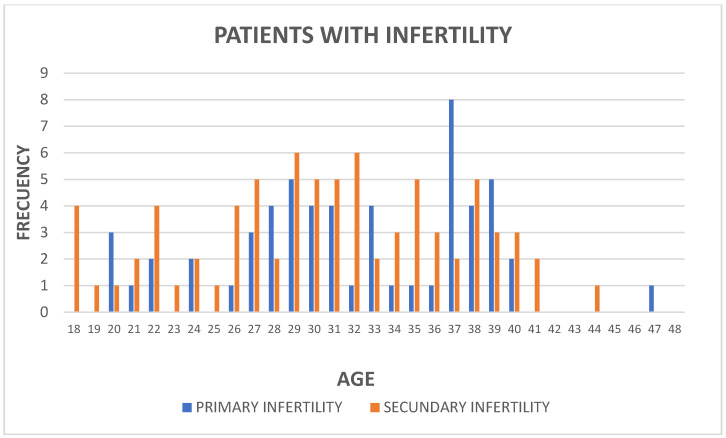
The histogram shows the age distribution of patients with primary infertility (blue bar) and secondary infertility (orange bar). Ages range from 18 to 48 years. For primary infertility, a peak is observed at 37 years, with patients predominantly distributed between 25 and 40 years and lower representation at the extremes of the range. For secondary infertility, the distribution is more uniform, with a peak around 29 years, consistent frequencies between 26 and 40 years, and some dispersion toward older ages (up to 45 years).

**Figure 2 microorganisms-13-00188-f002:**
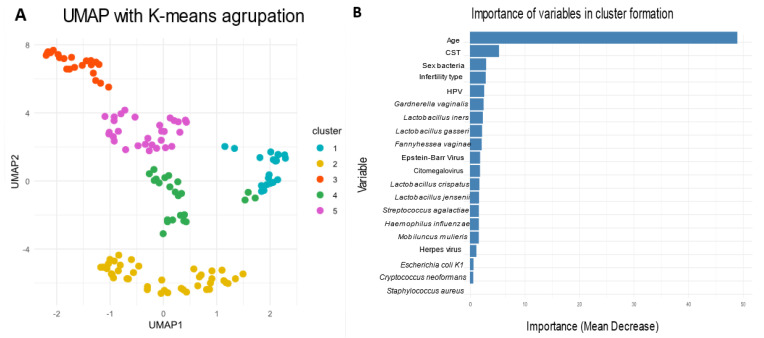
The vaginal microbiome community state types. (**A**) Cluster formation analysis. Five clusters were generated from all variables across all samples using the K-Means method. (**B**) Key variables impacting cluster formation. A Random Forest analysis was conducted to determine the variables that contribute to the groupings, with the most significant variables being age, microbial community type (CST), the presence of sexually transmitted bacteria, type of infertility, and infection with the human papillomavirus (HPV).

**Figure 3 microorganisms-13-00188-f003:**
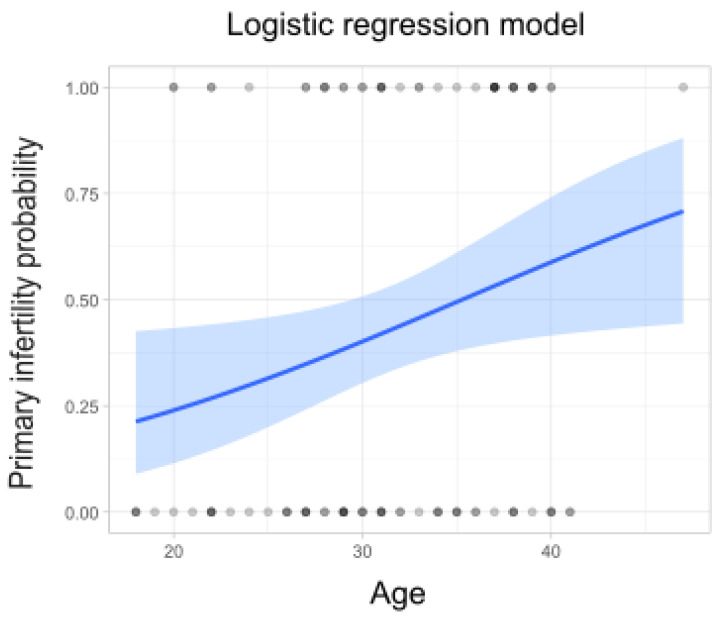
Relationship between age and type of infertility. Logistic regression models were constructed to evaluate the potential relationships between these factors. A positive relationship can be observed between increasing age and the likelihood of primary infertility.

**Figure 4 microorganisms-13-00188-f004:**
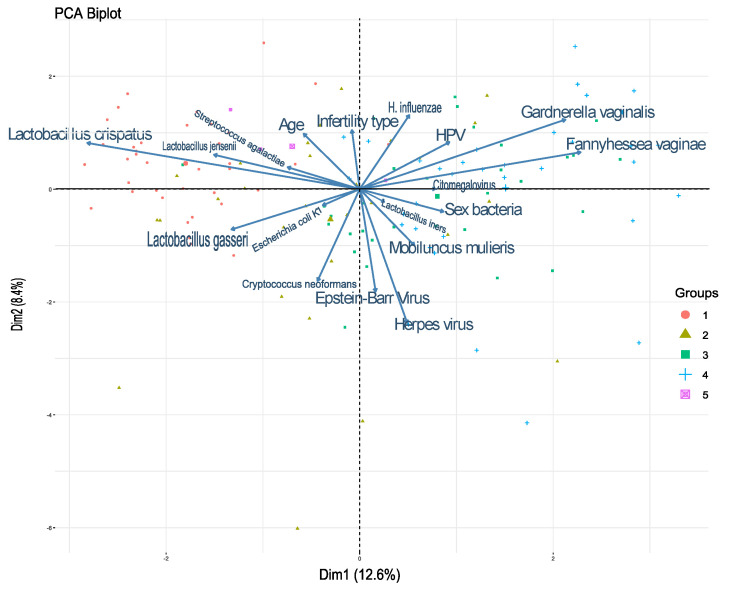
Correlations between bacterial species and viruses in different community state types. Principal component analysis (PCA) of community group types. The graph suggests correlations between the variables: smaller angles indicate a positive correlation, larger angles indicate a negative correlation, and 90° angles indicate no correlation between the characteristics. The distance between points represents their similarity: close points reflect similar profiles, while points that are distant exhibit dissimilar profiles.

## Data Availability

The original contributions presented in this study are included in the article/Appendix A. Further inquiries can be directed to the corresponding authors.

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
