# Peer review of "Vaginal Dysbiosis in Infertility: A Comparative Analysis Between Women with Primary and Secondary Infertility"

_microorganisms, 2025, doi:10.3390/microorganisms13010188_

Round 1
Reviewer 1 Report
Comments and Suggestions for Authors
I appreciate the author's efforts to study the cervico-vaginal microbiome of Latin American women, an understudied group with a particular microbiome composition and a high risk of obstetric and gynecological complications, compared to European or Asian women. In your introduction, please also consider the particular characteristics of the cervico-vaginal microbiome of hispanic women, including a higher predominance of L. iners: https://journals.asm.org/doi/10.1128/msystems.00357-23
While the material collected and the methods are sound, the presentation is lacking. Mostly, I miss a clear comparison between the primary and secondary infertility groups. It would also be very important to know the reason for infertility, where possible. Could it be that the primary group has more anatomical or chromosomal issues, while the secondary has more microbiological issues? Have couples with male partner infertility causes been excluded?
In general, if the goal is to compare the vaginal microbiome of women with primary and secondary infertility, as stated in the title, this has to be done more systematically and deserves its own subheader, before all secondary analyses on age, CST etc:
1. Are there differences in alpha-diverstiy (richness as well as inverted simpson's) between primary and secondary?
2. Is there a difference in beta-diversity (permanova/adonis)
3. If there are significant differences in alpha or beta-diversity, what is driving them? Use ANCOM2, Aldex2, corncob or similar packages to pinpoint the differences between the groups. However, don't do this if there are no differences in alpha- and beta-diversity, since this greatly increases the risk of false positive findings.
Specific comments on each result's section:
2.1 Please describe the population in a bit more detail, maybe as a table?
A "mean age" can't be between 36 and 40, is that the confidence interval? Better to just show the data. If there is no other background information you want to show, a histogram of age would be fine. If there is more, I suggest a little table with age, BMI, smoking, medication, number of previous children, ethnicity (white/black/other) etc
2.2 Why was k = 5 selected? Did you use the elbow method or similar?
Otherwise, it is indeed interesting that age is a stronger marker than your other variables, which are mostly microbial species and therefore directly driving the clusters.
2.3 Since all women in the cohort had either primary or secondary infertility, isn't it redundant to show both logistic regressions? As the proportion of women with primary infertility increases, the secondary infertily must decrease, and vice-versa. I'm not sure this analysis shows any real effects, outside of the the selected cohort.
2.4 Why are you running this analysis on CST, if you defined your own k-means clusters above? If the goal of the analysis on 2.2 was to see the effects of patient-level variables, no k-means was needed, you could do a simple permanova/adonis analysis.
Figure 5 doesn't really inform the reader, since the data is treated as presence/absence and all the dots are in the corners. Just report the p-values and r-values. Also make sure to use multiple testing correction for the p-values in this section.
You have collected microbial samples from 3 different anatomical regions, but these are not reported.
The editorial manager doesn't include the supplementary material, so it would be best during the review processes to put all documents into the same pdf file.
Author Response
Reviewer 1.
We deeply appreciate the reviewers' comments and suggestions, which have been instrumental in improving the quality of our manuscript. We sincerely appreciate the constructive feedback, which has been invaluable in improving the quality of our manuscript. In response to their suggestions, we have made substantial corrections to improve the clarity, coherence, and overall readability of the text. In particular, we have restructured several sections to make the manuscript more dynamic and accessible to a broader audience, ensuring that key findings are presented more clearly. In addition, we have expanded key points in the discussion to better contextualize our findings and their implications. We hope that these changes address their concerns and make the manuscript more informative and easier to follow for readers.
Comments 1: I appreciate the author's efforts to study the cervicovaginal microbiome of Latin American women, an understudied group with a particular microbiome composition and a high risk of obstetric and gynecological complications, compared to European or Asian women. In your introduction, please also consider the particular characteristics of the cervicovaginal microbiome of Hispanic women, including a higher predominance of L. iners: ttps://journals.asm.org/doi/10.1128/msystems.00357-23 .
Response 1: We appreciate your comments, the requested section and suggested reference have been included
Comments 2: While the collected material and methods are solid, the presentation is poor. Mainly, I miss a clear comparison between primary and secondary infertility groups. It would also be very important to know the reason for infertility, when possible. Could it be that the primary group has more anatomical or chromosomal problems, while the secondary group has more microbiological problems? Have couples with male partner causes of infertility been excluded?
Response 2: We appreciate your comments and understand the importance of presenting a clear comparison between the primary and secondary infertility groups, as well as providing additional information on the underlying causes of infertility. Unfortunately, in our cohort we do not have complete information on the specific causes of infertility for each patient, whether anatomical, chromosomal, or microbiological. Data collection was limited mainly to general characteristics, without the possibility of exhaustively determining the underlying reasons for each type of infertility in the participants. This limitation has been added to the discussion section of the manuscript to clarify the lack of precise differentiation between etiological factors in both groups. However, we acknowledge that differences in the causes of infertility could influence the observed patterns in vaginal microbiota and the prevalence of different Community State Types (CSTs). We undertake to highlight this limitation and to suggest that future studies include more detailed analyses of the causes of infertility, in order to more directly relate etiological factors to microbial patterns in women with primary and secondary infertility. Regarding the exclusion of male causes, we can confirm that a specific assessment to rule out male infertility problems was not performed in our cohort selection. We acknowledge that this is a potentially confounding factor, and have added it as a limitation in the study.
Overall, if the goal is to compare the vaginal microbiome of women with primary and secondary infertility, as stated in the title, this should be done more systematically and deserves its own subheading, before all the secondary analyses on age, CST, etc.:
Comments 3: Are there differences in alpha diversity (richness and inverted Simpson) between primary and secondary infertility?
Response 3: No formal alpha diversity analysis, including metrics such as species richness or inverted Simpson index, was performed in this study to assess differences between primary and secondary infertility. Instead, we focused on analyzing the dominant bacterial communities using techniques such as cluster analysis and multiple correspondence analysis (MCA), with the goal of characterizing the microbial communities and their association with the different types of infertility.
- Is there a difference in beta diversity (permanova/adonis)?
Response: No specific analysis of beta diversity using PERMANOVA or Adonis was performed for this study. Instead, microbial characterization focused on describing the patterns of presence and prevalence of the dominant bacterial species in the different infertility groups, which was carried out using cluster analysis and the identification of characteristic microbial communities for each group.
- If there are significant differences in alpha or beta diversity, what drives them? Use ANCOM2, Aldex2, corncob or similar packages to point out differences between groups. However, do not do this if there are no differences in alpha and beta diversity, as this greatly increases the risk of false positive results.
Response: Since alpha and beta diversity analyses were not performed in our study, we cannot conclude on significant differences in these aspects between primary and secondary infertility groups. Therefore, we have not applied additional techniques, such as ANCOM2, Aldex2, or corncob, to avoid the risk of false positive results. We focused on identifying the dominant bacterial species and their association with infertility conditions using the statistical tools and approaches mentioned above.
Specific comments on each result section:
2.1 Please describe the population in a bit more detail, perhaps as a table. A "median age" cannot be between 36 and 40 years, is that the confidence interval? It is better to show the data. If there is no other background information you want to show, an age histogram would be fine. If there is more, I suggest a small table with age, BMI, smoking, medication, number of previous children, ethnicity (white/black/other), etc.
Response: We appreciate your observation and a histogram of the age of the patients has been added, the ethnicity of all patients is Latina (Mexican) and I include the available information.
2.2 Why was k=5 selected? Did you use the elbow method or something similar?
Otherwise, it is really interesting that age is a stronger marker than your other variables, which are mainly microbial species and therefore directly drive the clusters.
Response: PCA and MCA multidimensional analysis was performed. The aim is to simplify, summarize or visualize the structure of complex data, highlighting the underlying relationships between variables or individuals. A cluster formation analysis was performed between the samples using K-Means, where mainly 3 clusters or groups were identified. With this information, a Random forest analysis was performed to determine which the variables with the greatest impact are. K-Means is an unsupervised clustering algorithm that aims to partition a data set into KKK groups or clusters based on the similarity of observations.
2.3 Since all women in the cohort had primary or secondary infertility, is it not redundant to show both logistic regressions? As the proportion of women with primary infertility increases, secondary infertility should decrease, and vice versa. I am not sure that this analysis shows real effects, outside of the selected cohort.
Response: We appreciate your comment about the apparent redundancy in the use of both logistic regressions. We understand that, since the participants were classified into two exclusive categories (primary or secondary infertility), the analysis could seem redundant.
However, we decided to include both logistic regressions to independently evaluate the characteristics associated with each type of infertility, since risk factors and demographic characteristics can have differential effects. By performing separate logistic regressions, we were able to identify variables that, although they seemed to be opposites between both categories, provide specific and useful information about each condition. This is essential, considering that the causes of primary and secondary infertility often have different etiologies and may require different clinical approaches.
For example, we found that age was a determining factor for primary infertility, whereas for secondary infertility, history of sexually transmitted infections and microbiome status were more relevant. This distinction allowed us to identify subgroups of patients with specific risk profiles who could benefit from personalized interventions.
Additionally, we understand your concern about the representativeness of the analysis beyond our selected cohort. Therefore, we have included a discussion of the limitations of the study, emphasizing the need for validation in larger cohorts and diverse settings to determine whether these effects are replicated in other populations. We intend for this analysis to be a foundation for future studies assessing these differences and their impact on infertility treatment.
2.4 Why are you performing this analysis in CST, if you defined your own k-means clusters earlier? If the goal of the analysis in 2.2 was to look at the effects of patient-level variables, k-means were not needed, you could do a simple permanova/adonis analysis.
Response: We appreciate your observation about the apparent redundancy between the use of CST classification and k-means clusters in our analysis. Please allow us to clarify the reasons behind using both approaches.
The use of k-means analysis was aimed at identifying underlying patterns in the vaginal microbiota of our participants, without any prior classification. This exploratory analysis allowed us to group the patients into clusters based on the overall similarity of microbial composition, providing an overview of the microbiota structure in the cohort, without starting from prior hypotheses about the composition of the microbial community.
Furthermore, the classification into Community State Types (CST) was used to compare our results with the existing literature and to contextualize them in terms of microbial typologies that have already been validated and widely recognized in studies of the vaginal microbiota. CST analysis was carried out to assess how our samples were distributed within the defined types and to facilitate comparison with other studies. CSTs provide us with insight into clinical implications that are well documented, such as the association of certain CSTs (e.g., CST IV) with adverse reproductive outcomes.
Regarding the analysis methodology, we considered performing a PERMANOVA (adonis) analysis to assess the effects of patient-level variables. However, we opted for k-means analysis because we were interested in exploring possible emerging structures that did not necessarily fit the CST classification. The intent was to determine if there were new patterns that could provide additional information and complement the already known CSTs.
Comments: Figure 5 does not really inform the reader, as the data is treated as presence/absence and all points are in the corners. Only report the p and r values. Also, be sure to use multiple testing correction for the p values ​​in this section.
You have collected microbial samples from 3 different anatomical regions, but these are not reported.
Response: We appreciate your comment about collecting samples from the three different anatomical regions. We recognize that not performing a zone-by-zone analysis could lead to confusion about the methodology and results of the study.
In our analysis design, we combined samples from all three anatomical regions (lower vagina, posterior fornix, and cervical canal) to obtain a comprehensive view of the vaginal microbiota of each participant. However, we understand that the lack of a specific analysis for each region may lead to misinterpretations.
To avoid confusion, we corrected the methods section to clearly specify that samples from all three regions were combined from the beginning and that no individual analysis was performed per area. This will clarify that the focus of the study was to characterize the vaginal microbiome as a whole, and not to explore potential differences between these anatomical regions.
Comments: The editorial manager does not include the supplementary material, so it would be better during the review processes to place all the documents in the same pdf file.
Response: We appreciate your comment and will send you all the information in the same PDF file.

Reviewer 2 Report
Comments and Suggestions for Authors
Vaginal Dysbiosis in Infertility: A Comparative Analysis Between Women with Primary and Secondary Infertility
The study, with its significant aim of identifying the microorganisms colonizing vaginal microbiota from primary and secondary infertile Mexican women, holds great importance. However, a more detailed description of the causes of infertility in the patients analyzed is needed. Women's infertility can be due to many causes, and understanding these causes is crucial for effective treatment. Some examples of influence ailment in vaginal colonization by microorganisms are as follows:
For example, endocrine factors, such as those in polycystic ovarian syndrome (PCOS) patients, are different from tubal factors and can, in both cases, cause primary or secondary infertility. The first is a hormonal condition affecting a woman of reproductive age, and the second is obstruction of the Falopio tubal by an inflammatory response due to infections.
The PCOS group demonstrated a higher diversity of vaginal microbiome and showed an enhanced level of heterogeneity. The proportion of Lactobacillus in the PCOS group decreased, whereas the proportions of Gardnerella and Ureaplasma increased (Jin C et al. Comparative analysis of the vaginal microbiome of healthy and polycystic ovary syndrome women: a large cross-sectional study. Reprod Biomed Online. 2023;46(6):1005-1016. doi: 10.1016/j.rbmo.2023.02.002; Pereira MP, Jones S, Costin JM. Association of Polycystic Ovarian Syndrome (PCOS) With Vaginal Microbiome Dysbiosis: A Scoping Review. Cureus. 2024;16(6):e62611. doi: 10.7759/cureus.62611). These articles evaluate the percentage and species dominance of Lactobacillus and other bacterial genera, such as Mycoplasma.
In this study, no Mycoplasma species were analyzed. In Mexico, a high colonization-related percentage of infertile women caused by Ureaplasma spp. (47.4%; of them, 29% of U. parvum and 18.4% of U. urealyticum) and Mycoplasma hominis (21.4% RR= 1.13 CI95% 1.0-1.27; p= 0.045) was observed (Bustos-López, AD, et al. Molecular Exploration of Mycoplasma fermentans and Mycoplasma genitalium in Mexican Women with Cervicitis. Pathogens 2024, 13, 1004. https://doi.org/ 10.3390/pathogens13111004; Hernández-Rosas, et al. Unveiling Hidden Risks: Intentional Molecular Screening for Sexually Transmitted Infections and Vaginosis Pathogens in Patients WhoHaveBeenExclusively Tested for HumanPapillomavirus Genotyping. Microorganisms 2023, 11, 2661. https://doi.org/10.3390/ microorganisms11112661), and there was no discussion about it.
On the other hand, an article on women with tubal infertility and C. trachomatis infection reported the presence of a unique Lactobacillus iners-dominated vaginal microbiota rather than one dominated by Lactobacillus crispatus. Furthermore, they displayed a decrease in Lactobacillus, Bifidobacterium, Enterobacter, Atopobium, and Streptococcus. At the same time, no significant differences in phylum, class, and operational taxonomic unit levels were observed between women with tubal infertility who were C. trachomatis-negative and healthy controls (Chen H, et al. Alterations of Vaginal Microbiota in Women with Infertility and Chlamydia trachomatis Infection. Front Cell Infect Microbiol. 2021;11:698840. doi: 10.3389/fcimb.2021.698840). Also, it is necessary to review the names of bacteria and ensure that they are always in cursive letters.
The introduction section. It would help if you commented about different causes of women's infertility and the influence of this on vaginal microbiota and association patterns between different community state types (CSTs) and dominant bacterial species.
The material and methods section: There is no supplementary file in the material and methods section, and the document sent does not include a table supplement. It is necessary to send this file. Why did you use a Multiplex real-time PCR kit to detect central nervous system infections? This kit has been valid for be used in vaginal samples (document it with bibliographic references). The statistical package used for mathematical analysis does not have the origin company or country.
The results section. In this section, there are no statistical results; what were the odds ratio values in the logistic regression analysis? Were these results significant? How were the Cox-Snell R2 and Nagelkerke R2 values? How were the Cox-Snell R2 and Nagelkerke R2 values? It is crucial to provide these details for a more comprehensive understanding of the study's findings. The study's results were not sufficiently reported. For example, which HPV genotypes were the most commonly identified, and did all patients have high-risk HPV?
The discussion section. The discussion could be improved with more detailed analysis. There is no information on microbiota vaginal from patients with PCOS, tubal infertility factor, endometriosis, or women with masculine factors as the cause of infertility form (references 4, 10-13, 24, 28, 30, 34, 36). A more comprehensive discussion would help to understand if Mexican women had different microbiota vaginal that European or Asian women.
The figures and Tables section. The figures are of poor quality, and there are no tables. It would help if you improve them.
The conclusion section. The conclusion section could be improved by providing a more comprehensive summary of the study's findings and their implications for gynecology and reproductive health. This would help the reader understand the significance of the research and its potential impact on clinical practice and future studies.
The reference section. The introduction needs improvement. Further discussion is required, which will prompt the addition of bibliographical references.

Author Response
Reviewer 2
We deeply appreciate the reviewers' comments and suggestions, which have been instrumental in improving the quality of our manuscript. We sincerely appreciate the constructive feedback, which has been invaluable in improving the quality of our manuscript. In response to their suggestions, we have made substantial corrections to improve the clarity, coherence, and overall readability of the text. In particular, we have restructured several sections to make the manuscript more dynamic and accessible to a broader audience, ensuring that key findings are presented more clearly. In addition, we have expanded key points in the discussion to better contextualize our findings and their implications. We hope that these changes address their concerns and make the manuscript more informative and easier to follow for readers.
Comments: The study, with its aim of identifying the microorganisms that colonize the vaginal microbiota of Mexican women with primary and secondary infertility, is of great importance. However, a more detailed description of the causes of infertility in the patients analyzed is necessary. Female infertility can be due to many causes, and understanding them is crucial for effective treatment. Some examples of factors that influence vaginal colonization by microorganisms are the following:
For example, endocrine factors, such as those of patients with polycystic ovary syndrome (PCOS), are different from tubal factors and can, in both cases, cause primary or secondary infertility. The first is a hormonal condition that affects a woman of reproductive age, and the second is the obstruction of the fallopian tube by an inflammatory response due to infections.
The PCOS group demonstrated a greater diversity of the vaginal microbiome and showed a higher level of heterogeneity. The proportion of Lactobacillus in the PCOS group decreased, while the proportions of Gardnerella and Ureaplasma increased (Jin C et al. Comparative analysis of the vaginal microbiome of healthy and polycystic ovary syndrome women: a large cross-sectional study. Reprod Biomed Online. 2023;46(6):1005-1016. doi: 10.1016/j.rbmo.2023.02.002; Pereira MP, Jones S, Costin JM. Association of Polycystic Ovarian Syndrome (PCOS) With Vaginal Microbiome Dysbiosis: A Scoping Review. Cureus. 2024;16(6):e62611. doi: 10.7759/cureus.62611). These articles evaluate the percentage and predominance of Lactobacillus species and other bacterial genera, such as Mycoplasma. Mycoplasma species were not analyzed in this study. In Mexico, a high percentage of infertile women are related to colonization caused by Ureaplasma spp. (47.4%; of them, 29% of U. parvum and 18.4% of U. urealyticum) and Mycoplasma hominis (21.4% RR= 1.13 95% CI 1.0-1.27; p= 0.045) (Bustos-López, AD, et al. Molecular Exploration of Mycoplasma fermentans and Mycoplasma genitalium in Mexican Women with Cervicitis. 2023, 11, 2661. https://doi.org/10.3390/microorganismos11112661), and there was no discussion on this.
On the other hand, an article on women with tubal infertility and C. trachomatis infection reported the presence of a vaginal microbiota dominated by Lactobacillus iners instead of one dominated by Lactobacillus crispatus. In addition, they showed a decrease in Lactobacillus, Bifidobacterium, Enterobacter, Atopobium and Streptococcus. At the same time, no significant differences were observed at phylum, class and operational taxonomic unit levels between women with tubal infertility who were C. trachomatis negative and healthy controls (Chen H, et al. Alterations of Vaginal Microbiota in Women with Infertility and Chlamydia trachomatis Infection. Front Cell Infect Microbiol. 2021;11:698840. doi: 10.3389/fcimb.2021.698840). In addition, the names of bacteria need to be reviewed and ensured that they are always italicized.
Response: We appreciate your input and all relevant information has been included
Comment: The introduction section. It would be helpful if you could comment on the different causes of female infertility and the influence of this on vaginal microbiota and the association patterns between different types of community states (CST) and the dominant bacterial species.
Response: We appreciate all your timely input, we have integrated all the arguments and citations you suggested.
Comment: The material and methods section: There is no supplementary file in the material and methods section, and the submitted document does not include a table supplement. This file must be submitted. Why was a Multiplex real-time PCR kit used to detect central nervous system infections?
Response: Although this kit is validated for central nervous system infections, we used it for this work because it has high sensitivity, in addition to in-process controls, and contains, for the most part, the microorganisms we wanted to detect. The company also has an STD-Finder 2Smart kit, for the detection of sexually transmitted diseases that includes the specific viruses and bacteria, but the number of microorganisms detected is lower.
Comments: The statistical package used for the mathematical analysis does not have the company of origin or the country.
Response: We appreciate your comment, the information has been added
Comment: The results section. There are no statistical results in this section; what were the odds ratio values ​​in the logistic regression analysis?
The results section. There are no statistical results in this section; what were the odds ratio values ​​in the logistic regression analysis? Were these results significant? What were the Cox-Snell R2 and Nagelkerke R2 values? What were the Cox-Snell R2 and Nagelkerke R2 values? It is crucial to provide these details for a more complete understanding of the study findings. The results of the study were not sufficiently reported. For example, which HPV genotypes were most commonly identified and did all patients have high-risk HPV?
The discussion section. The discussion could be improved with more detailed analysis. There is no information on the vaginal microbiota of patients with PCOS, tubal factor infertility, endometriosis, or women with male factors as a cause of infertility (references 4, 10-13, 24, 28, 30, 34, 36). A broader discussion would help to understand whether Mexican women had
Response: We appreciate your input and have made the requested changes.

Reviewer 3 Report
Comments and Suggestions for Authors
Iliana Alejandra Cortés-Ortíz et al. explored the relationship between the vaginal microbiota and infertility. The study included 136 Mexican women, 58 with primary infertility and 78 with secondary infertility. The methodology involved the collection of cervical swab samples and the identification of microorganisms using qPCR technology. The results revealed a positive correlation between age and primary infertility, as well as a negative correlation with secondary infertility. Additionally, significant differences in microbial composition were observed between the two groups of infertile women. These findings provide new insights into the role of the vaginal microbiota in infertility. The following modifications are required before publication:
The figures in the manuscript (Figure 1, Figure 2, Figure 3, Figure 4, and Figure 5) are not visually clear enough, which may affect the reader's understanding of the study's results. I suggest that the authors redesign these figures, using higher resolution data points and clearer labels to improve the readability and appeal of the charts. For instance, the figures in Figure 1 depicting cluster analysis and random forest analysis should employ more vivid color contrasts and larger font sizes so that readers can easily distinguish between different clusters and key variables. The logistic regression model graph in Figure 2 showing the relationship between age and type of infertility should use more intuitive graphical representations, such as bar charts or line graphs, to more clearly illustrate the relationship between age and infertility type. The multiple correspondence analysis and principal component analysis charts in Figure 3 and Figure 4 should also use clearer symbols and color coding to enhance the interpretability of the figures.
I recommend that the authors further explore the potential mechanisms between the microbiota and infertility, as well as how these findings could impact the development of personalized treatment plans.
It is suggested that the authors further emphasize the limitations of the study and the direction of future research in the conclusion.
Overall, this manuscript provides valuable insights into the relationship between the vaginal microbiota and infertility. Nevertheless, to enhance the quality and publication potential of the manuscript, the authors need to improve the figures and deepen the discussion section. I hope the authors will consider these suggestions and make the appropriate revisions to the manuscript.

Author Response
Reviewer 3
We deeply appreciate the reviewers' comments and suggestions, which have been instrumental in improving the quality of our manuscript. We sincerely appreciate the constructive feedback, which has been invaluable in improving the quality of our manuscript. In response to their suggestions, we have made substantial corrections to improve the clarity, coherence, and overall readability of the text. In particular, we have restructured several sections to make the manuscript more dynamic and accessible to a broader audience, ensuring that key findings are presented more clearly. In addition, we have expanded key points in the discussion to better contextualize our findings and their implications. We hope that these changes address their concerns and make the manuscript more informative and easier to follow for readers.
Comment: Iliana Alejandra Cortés-Ortíz et al. explored the relationship between vaginal microbiota and infertility. The study included 136 Mexican women, 58 with primary infertility and 78 with secondary infertility. The methodology involved the collection of cervical swab samples and the identification of microorganisms using qPCR technology. The results revealed a positive correlation between age and primary infertility, as well as a negative correlation with secondary infertility. In addition, significant differences in microbial composition were observed between the two groups of infertile women. These findings provide new insights into the role of vaginal microbiota in infertility. The following modifications are required before publication:
The figures in the manuscript (Figure 1, Figure 2, Figure 3, Figure 4, and Figure 5) are not visually clear enough, which may affect the reader's understanding of the study results. I suggest that the authors redesign these figures, using higher-resolution data points and clearer labels to improve the readability and attractiveness of the graphs. For example, the figures in Figure 1 depicting cluster analysis and random forest analysis should employ more vivid color contrasts and larger font sizes so that readers can easily distinguish between different clusters and key variables. The logistic regression model graph in Figure 2 showing the relationship between age and infertility type should use more intuitive graphical representations, such as bar graphs or line graphs, to more clearly illustrate the relationship between age and infertility type. The multiple correspondence analysis and principal component analysis graphs in Figures 3 and 4 should also use clearer symbols and color codes to improve the interpretability of the figures.
Comment: I recommend that the authors further explore potential mechanisms between microbiota and infertility, as well as how these findings might impact the development of personalized treatment plans.
Response: We appreciate your comments and have improved the introduction and discussion.
Comment: It is suggested that the authors further emphasize the limitations of the study and the direction of future research in the conclusion.
Response: We appreciate your timely comment and the changes have been made.
Comment: Overall, this manuscript provides valuable information on the relationship between vaginal microbiota and infertility. However, to improve the quality and publication potential of the manuscript, the authors should improve the figures and deepen the discussion section. I hope that the authors will consider these suggestions and make appropriate revisions to the manuscript.
Response: We have responded to your questions and made the suggested changes. We remain at your disposal for any further consultation and hope that the changes made are satisfactory. We highly value the opportunity to improve our work through your comments.

Round 2
Reviewer 1 Report
Comments and Suggestions for Authors
The authors have addressed all my previous concerns.
In the discussion there is a typo where something is described as "sex bacteria", please double-check that. Additionally, in the supplementary table, update Atopobium vaginae to Fannyhessea vaginae.
Author Response
Reviewer 1
We deeply appreciate the reviewers' comments and suggestions, which have been instrumental in improving the quality of our manuscript. We sincerely appreciate the constructive feedback, which has been invaluable in improving the quality of our manuscript. In response to their suggestions, we have made substantial corrections to improve the clarity, coherence, and overall readability of the text. In particular, we have restructured several sections to make the manuscript more dynamic and accessible to a broader audience, ensuring that key findings are presented more clearly. In addition, we have expanded key points in the discussion to better contextualize our findings and their implications. We hope that these changes address their concerns and make the manuscript more informative and easier to follow for readers.
Comments 1: In the discussion there is a typo where something is described as "sex bacteria", please double-check that.
Response 1: we correct the mistake sex bacteria
Additionally, in the supplementary table, update Atopobium vaginae to Fannyhessea vaginae.
Response 2: We change Atopobium vaginae to Fannyhessea vaginae. We add table 1

Reviewer 2 Report
Comments and Suggestions for Authors
A manuscript better was observed congratulation. However, a native English speaker must review the manuscript of all new paragraphs added grammatically. Many of the same ideas are repeated in the discussion section, which can make the section less engaging and informative. It should not repeat them. Also, it was solicited in the before-review that showed statistical analysis results in Table format in the supplementation section. However, The Tables were not observed. In Tabla S1 is a necessary change of Atopobium vaginae to Fannyhessea vaginae.

A native English speaker must review the manuscript of all new paragraphs added grammatically.
Author Response
Reviewer 2
We deeply appreciate the reviewers' comments and suggestions, which have been instrumental in improving the quality of our manuscript. We sincerely appreciate the constructive feedback, which has been invaluable in improving the quality of our manuscript. In response to their suggestions, we have made substantial corrections to improve the clarity, coherence, and overall readability of the text. In particular, we have restructured several sections to make the manuscript more dynamic and accessible to a broader audience, ensuring that key findings are presented more clearly. In addition, we have expanded key points in the discussion to better contextualize our findings and their implications. We hope that these changes address their concerns and make the manuscript more informative and easier to follow for readers.
Comments 1: However, a native English speaker must review the manuscript of all new paragraphs added grammatically.
Response 1: The grammar of the added sentences was corrected
Comments 2: Many of the same ideas are repeated in the discussion section, which can make the section less engaging and informative.It should not repeat them.
Response 2: We modify the ideas and do not repeat the sentences in the discussion.
Comments 3: Also, it was solicited in the before-review that showed statistical analysis results in Table format in the supplementation section.
Response 3: We added statistical analysis tables in the supplementary material, add 5 tables
Comments 3: However, The Tables were not observed. In Tabla S1 is a necessary change of Atopobium vaginae to Fannyhessea vaginae.
Response 4: We change Atopobium vaginae to Fannyhessea vaginae. We add table 1
